# Association between a common immunoglobulin heavy chain allele and rheumatic heart disease risk in Oceania

Tom Parks[1], Mariana M. Mirabel[2], Joseph Kado[3,4], Kathryn Auckland[1], Jaroslaw Nowak[5], Anna Rautanen[1], Alexander J. Mentzer[1], Eloi Marijon[2,6], Xavier Jouven[2,6], Mai Ling Perman[4], Tuliana Cua[7], John K. Kauwe[8], John B. Allen[8], Henry Taylor[9], Kathryn J. Robson[10], Charlotte M. Deane[5], Andrew C. Steer[11,12,*], Adrian V.S. Hill[1,*] & for the Pacific Islands Rheumatic Heart Disease Genetics Network[†]

The indigenous populations of the South Pacific experience a high burden of rheumatic heart disease (RHD). Here we report a genome-wide association study (GWAS) of RHD susceptibility in 2,852 individuals recruited in eight Oceanian countries. Stratifying by ancestry, we analysed genotyped and imputed variants in Melanesians (607 cases and 1,229 controls) before follow-up of suggestive loci in three further ancestral groups: Polynesians, South Asians and Mixed or other populations (totalling 399 cases and 617 controls). We identify a novel susceptibility signal in the immunoglobulin heavy chain (IGH) locus centring on a haplotype of nonsynonymous variants in the *IGHV4-61* gene segment corresponding to the *IGHV4-61*02* allele. We show each copy of *IGHV4-61*02* is associated with a 1.4-fold increase in the risk of RHD (odds ratio 1.43, 95% confidence intervals 1.27–1.61, $P = 4.1 \times 10^{-9}$). These findings provide new insight into the role of germline variation in the IGH locus in disease susceptibility.

[1] Wellcome Trust Centre for Human Genetics, University of Oxford, Roosevelt Drive, Oxford OX3 7BN, UK. [2] Paris Centre de Recherche Cardiovasculaire, Institut National de la Santé et de la Recherche Médicale, Hôpital Européen Georges Pompidou, 56, rue Leblanc, 75908 Paris, France. [3] Department of Paediatrics, Ministry of Health and Medical Services, Colonial War Memorial Hospital, Brown Street, Suva, Fiji. [4] College of Medicine, Nursing & Health Sciences, Fiji National University, Brown Street, Suva, Fiji. [5] Department of Statistics, University of Oxford, Peter Medawar Building for Pathogen Research, Oxford OX1 3S, UK. [6] Faculté de Médecine Paris Descartes, Université Paris Descartes, 15, rue de l'école de medicine, 75006 Paris, France. [7] Rheumatic Heart Disease Control Programme, Ministry of Health and Medical Services, Colonial War Memorial Hospital, Brown Street, Suva, Fiji. [8] College of Life Sciences, Brigham Young University, 4146 Life Sciences Building, Provo, Utah 84602, USA. [9] Rheumatic Heart Disease Control Programme, Samoa Ministry of Health, Moto'otua, Ifiifi Street, Apia, Samoa. [10] MRC Weatherall Institute of Molecular Medicine, University of Oxford, John Radcliffe Hospital, Headington, Oxford OX3 9DS, UK. [11] Centre for International Child Health, University of Melbourne, 50 Flemington Road, Parkville, Victoria 3052, Australia. [12] Murdoch Children's Research Institute, 50 Flemington Road, Parkville, Victoria 3052, Australia. * These authors contributed equally to this work. Correspondence and requests for materials should be addressed to A.C.S. (email: andrew.steer@rch.org.au) or to A.V.S.H. (email: adrian.hill@ndm.ox.ac.uk).
[†] A full list of consortium members appears at the end of the paper.

Rheumatic heart disease (RHD) is the chronic consequence of an aberrant immune response to *Streptococcus pyogenes* (also termed group A streptococcus (GAS)), a process that leads to scarring and dysfunction of heart valves. Previously, a major public health concern in Europe and the United States, the disease remains a prominent cause of death, heart failure and stroke among young and middle-aged adults in developing countries[1]. Although reliable data remain scarce, it is likely the disease affects at least 16 million individuals worldwide, causing an estimated 300,000 premature deaths each year[2]; however, relative to its global impact, RHD has been largely neglected by researchers and funders alike[3]. Consequently, there has been limited progress towards understanding pathogenesis that has hampered efforts in disease control and development of novel therapies and an effective vaccine[4].

Host genetic susceptibility is one compelling feature of the disease that awaits rigorous investigation. For over a century, clinicians have noted the strong familial propensity of acute rheumatic fever (ARF)[5], and it was recently estimated on the basis of twin studies dating back to the 1930s that monozygotic twins have sixfold greater concordance than dizygotic twins[6]. Moreover, even in highly endemic settings where childhood GAS infections are ubiquitous, only a minority of the population develop ARF or RHD during their lifetime (up to 5–6%), and this may indicate that the disease develops only in those who are genetically predisposed[7]. Despite this, efforts to delineate host genetic susceptibility have so far been limited to a number of small candidate gene studies—many focused on the HLA locus— the results of which have been inconsistent and largely inconclusive[8].

Here we report a genome-wide association study (GWAS) of RHD susceptibility in the endemic settings of Oceania, where the disease remains a leading cause of premature death and disability[9]. We identify a novel susceptibility signal in the immunoglobulin heavy chain (IGH) locus centring on a haplotype of nonsynonymous variants in the *IGHV4-61* gene segment corresponding to the *IGHV4-61*02* allele. Set in populations hitherto largely overlooked by genetics research, to the best of our knowledge, our study is the first GWAS of RHD, providing much needed insight into the pathogenesis of this devastating disease. Additionally, as the only study from the GWAS era that we are aware of linking germline coding variants in the IGH locus to disease susceptibility, our study suggests further consideration should be given to the role of IGH polymorphism in autoimmune disease.

## Results

**Genome-wide association analysis.** Our study was undertaken using a collection of 3,412 DNA samples from individuals recruited in eight Oceanian countries established by the Pacific Islands RHD Genetics Network (Fig. 1a). For this analysis we successfully genotyped 3,234 individuals at 239,990 variants using the Illumina HumanCore platform (Supplementary Fig. 1b,c). To supplement the genotype data, we imputed genotypes of variants falling between those assayed directly. However, owing to the absence of Oceanian populations from current reference panels, we undertook low-coverage whole-genome sequencing of 64 Melanesian individuals recruited in New Caledonia (Supplementary Fig. 2a–c). As suggested previously[10], we phased 9,489,051 variants identified through sequencing (13.0% of which were novel) onto a haplotype scaffold of 622,740 variants, ascertained by genotyping the same individuals and a further 64 individuals recruited in Fiji using the Illumina HumanOmniExpressExome platform, a higher density array. We then performed genome-wide imputation using the phased Oceanian sequenced data

(128 haplotypes) integrated with the phase 3 release from the 1000 Genomes Consortium (5,008 haplotypes). Testing the utility of the integrated panel, we found the mean sample concordance, a standard measure of imputation accuracy, improved by 4–5% in individuals of Oceanian ancestry as compared with imputation using the 1000 Genomes reference panel alone (Supplementary Fig. 2d).

The samples available to us were of diverse genetic ancestry reflecting not only their varied provenance but also underlying structure and admixture (Fig. 1). We chose first to focus on identifying susceptibility variants with consistent direction and magnitude of effects across the data set, not least because such trans-ancestral analysis can help fine-map causal variation[11]. We therefore used principal components analysis to assign individuals to one of four ancestral strata: Melanesian; Polynesian; Fijian Indian, that is, South Asian; Mixed or other (Supplementary Fig. 3a–d). Then, after pruning first- and second-degree relatedness, we performed case–control association tests within each strata, using linear mixed models (LMM) to minimize residual confounding due to residual structure (Supplementary Fig. 3e) and more distant relatedness (Supplementary Fig. 4b). Having performed a discovery analysis by LMM in the Melanesian strata ($\lambda = 1.06$; Supplementary Fig. 5a), we combined the resulting association statistics with those from LMM analyses from the remaining three strata ($\lambda = 1.00$–$1.02$; Supplementary Fig. 5b–d) using fixed effects (FE) inverse variance-weighted meta-analysis ($\lambda = 1.05$; Supplementary Fig. 5e) that is widely considered the first choice meta-analysis strategy for variant discovery[12].

Of the 24 independent signals at suggestive significance in the discovery analysis (Supplementary Fig. 6), only a signal located in the IGH locus on chromosome 14 showed evidence of replication (Fig. 2). Comprising 102 variants at genome-wide significance, of which two had been directly genotyped, the signal peaked at a single nucleotide polymorphism (SNP) located 6 kb upstream from the *IGHV4-61* gene segment (rs11846409, FE meta-analysis, $P = 3.6 \times 10^{-9}$; Supplementary Fig. 7a). This variant was imputed with certainty 97.5% (information (info.) metric 0.953) and was significantly associated with susceptibility in all four ancestral strata (LMM, $P = 1.7 \times 10^{-5}$ to $P = 0.037$). To fine-map this signal, we performed Bayesian trans-ancestral meta-analysis using genetic distance between the populations as a prior (Supplementary Fig. 7b)[11] and, as previously described, defined a set of 183 credible variants that was 99% likely to include the causal variant (Fig. 3a)[13]. Six of this set were annotated as coding of which five were located in the second exon of *IGHV4-61* (Supplementary Fig. 7c), all part of the previously defined *IGHV4-61*02* allele[14].

**Confirmation by Sanger sequencing.** To resolve the signal further, we undertook chain-termination ('Sanger') sequencing of a 473 base-pair segment of the second exon of *IGHV4-61* in a subset of the samples (Supplementary Fig. 8). Among the 339 sequenced individuals included in the association analyses we identified three common haplotypes (Supplementary Fig. 9), two known, matching the *IGHV4-61*01* and *IGHV4-61*02* alleles, as previously defined, and one novel, comprising a six base in-frame deletion and a nonsynonymous variant that converts the amino acid sequence of *IGHV4-61* to that of *IGHV4-59*, provisionally designated *IGHV4-61*09* (Supplementary Fig. 10). Although the complexity of the IGH locus makes it difficult to be certain, it seems most likely that this novel allele has been amplified from the *IGHV4-61* locus rather than the *IGHV4-59* locus because the sequence surrounding *IGHV4-61*09* matched the former better than the latter (Supplementary Note 1, Supplementary Fig. 11).

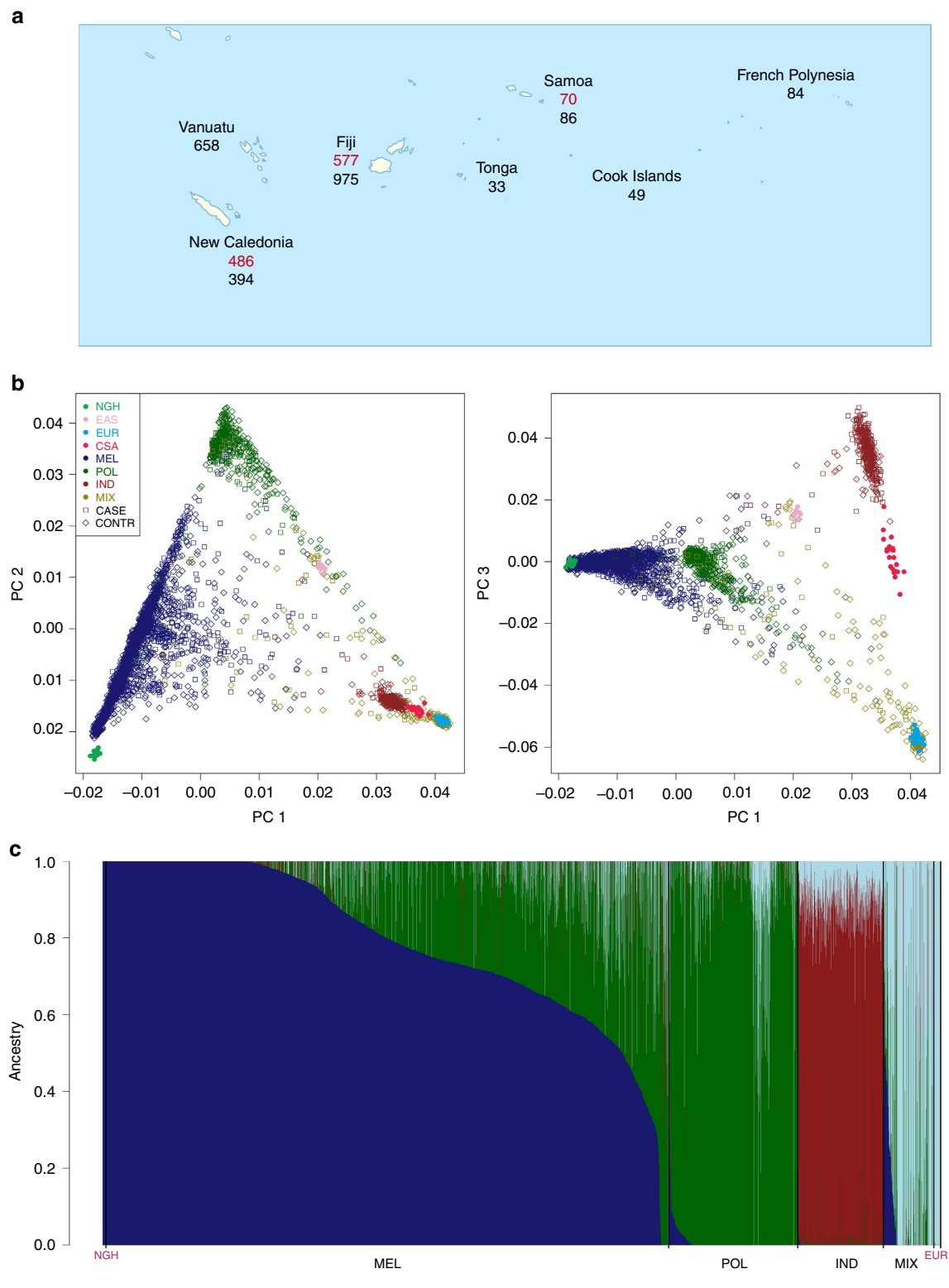

**Figure 1 | Oceanian study population.** (**a**) Approximate location where genotyped cases (red) and controls (black) were sampled. (**b**) Projection of the samples on to the first and second (left) and first and third (right) principal components (PCs) of genetic variation coloured by self-reported ancestry (MEL, Melanesians; POL, Polynesian; IND, Fijian Indian; MIX, Mixed and other) with cases indicated by empty squares and controls by empty diamonds. Selected samples from the Human Genome Diversity Project Panel (NGH, Papuan; EAS, South East Asian; EUR, European; CSA, Central South Asian) are superimposed for comparison and indicated by filled circles. (**c**) Estimates of admixture proportions from four source populations grouped by self-reported ancestry, with selected samples of Papuan and European ancestry shown at the far left and right, respectively, for comparison.

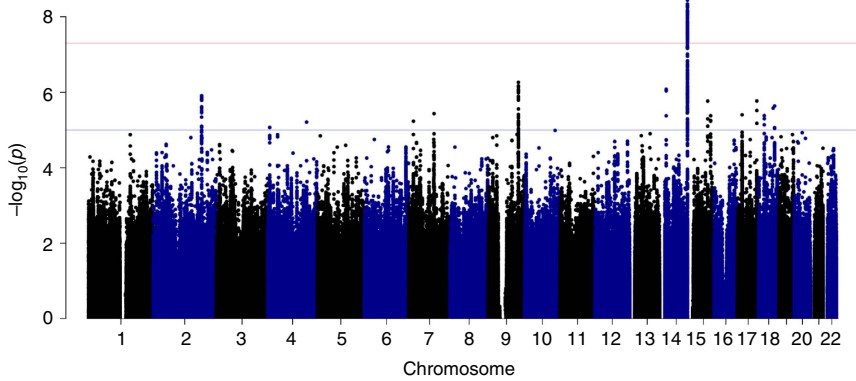

**Figure 2 | Genome-wide meta-analysis for RHD susceptibility.** For each variant, the negative common logarithm of the $P$ value from an inverse-variance weighted fixed-effects meta-analysis is plotted against genomic position. The blue horizontal line indicates suggestive significance (FE meta-analysis, $P = 10^{-5}$) and the red horizontal line indicates genome-wide significance (FE meta-analysis, $P = 5 \times 10^{-8}$).

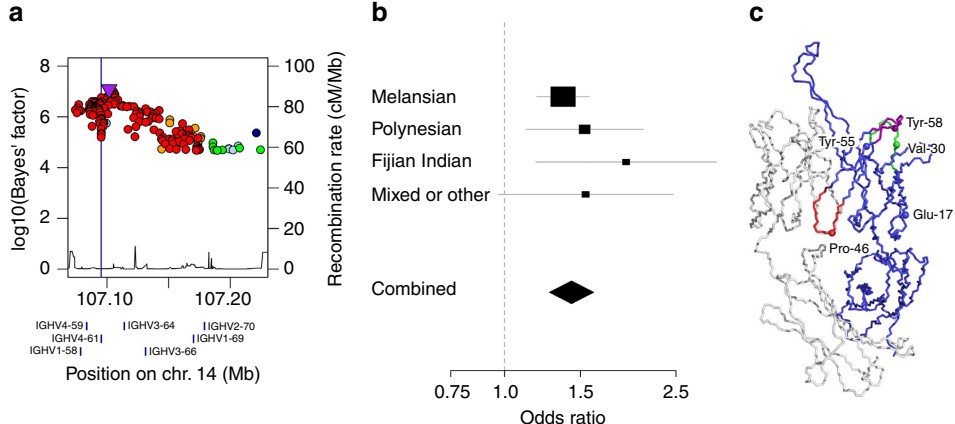

**Figure 3 | Association of the *IGHV4-61* locus with RHD susceptibility.** (**a**) For each variant in the 99% credible set, the common logarithm of the Bayes' factor is plotted against genomic position. Variants are coloured by linkage disequilibrium with the most associated variant averaged across the entire data set (estimated $r^2$: dark blue, 0–0.2; light blue, 0.2–0.4; green, 0.4–0.6; yellow, 0.6–0.8; red, 0.8–1.0). A vertical blue line indicates the position of the four nonsynonymous variants in *IGHV4-61* and locations of expressed IGH gene segments are indicated by blue rectangles below the x axis. (**b**) Forest plot for the *IGHV4-61\*02* allele under an additive genetic model with association statistics from LMM analysis in each strata combined by FE meta-analysis. Individual and combined odds ratio estimates with confidence intervals are shown on a logarithmic scale. (**c**) Structural model of an antibody that includes the *IGHV4-61* heavy variable domain (Protein Databank 4FQQ) showing both heavy (blue) and light (white) chains with both the first (CDR-H1, green) and second (CDR-H2, violet) heavy chain complementarity determining loops and the heavy chain interface framework loop (HIFL, red) highlighted. The positions that distinguish *IGHV4-61\*01* from *IGHV4-61\*02* are shown as spheres labelled with the amino acids found in *IGHV4-61\*01*.

When locally imputed into the wider data set, the *IGHV4-61\*02* allele was predicted far more accurately (certainty 97.0%, info. metric 0.935) than its component SNPs had been by genome-wide imputation (certainty 51.4–71.7%, info. metric 0.797–0.877). Using the locally imputed data, we found each copy of *IGHV4-61\*02*, which had minor allele frequency 24.9%, was associated with a 1.4-fold increased risk of disease (odds ratio 1.43, 95% confidence intervals 1.27–1.61, FE meta-analysis, $P = 4.1 \times 10^{-9}$; Table 1). This *IGHV4-61\*02* signal was very marginally weaker than that for the lead SNP from the genome-wide analysis (rs11846409, FE meta-analysis, $P = 3.6 \times 10^{-9}$), most likely reflecting residual uncertainty surrounding the imputed *IGHV4-61\*02* genotypes; however, in an analysis limited to the 339 sequenced individuals included in the association analyses, the signal for *IGHV4-61\*02* (LMM, $P = 0.041$) was stronger than that for rs11846409 (LMM, $P = 0.062$). Across the data set, the *IGHV4-61\*02* signal showed strikingly little heterogeneity between the ancestral strata (Cochran's $Q$ test, $P = 0.55$; Fig. 3b) and a broadly additive relationship between disease and genotype in each (Supplementary

Fig. 12a–d). Moreover, conditioned on *IGHV4-61\*02*, we found neither the aforementioned novel deletion haplotype (*IGHV4-61\*09*, FE meta-analysis, $P = 0.50$) nor other variants in the *IGHV4-61* locus ($\pm 250$ kb, FE meta-analysis, minimum $P = 0.045$) remained associated with disease. Furthermore, the association between *IGHV4-61\*02* and disease remained statistically significant across a variety of populations and subpopulations tested as sensitivity analyses (Table 1) including analyses limited to four subsets of case–control pairs matched by ancestry (FE meta-analysis $P = 4.1 \times 10^{-8}$; Supplementary Fig. 13a) and the three countries in which independent case–control studies had been undertaken (FE meta-analysis, $P = 8.6 \times 10^{-9}$; Supplementary Fig. 13b). Finally, in a supplemental analysis involving children recruited in Samoa with mild nondiagnostic valve abnormalities, borderline RHD or definite RHD, the latter two based on criteria published by the World Heart Federation[15], each compared with the Samoan controls used in the main analysis, we found the effect of *IGHV4-61\*02* strongly correlated with diagnostic certainty, there being nil, marginal and significant effect, respectively (Supplementary Fig. 13c).

**Table 1 | Association of the *IGHV4-61\*02* allele with RHD susceptibility by ancestry and country.**

| Grouping | Population | Subpopn | Subgroup | Cases N | Controls N | Effective N | Minor allele freq. Cases | Minor allele freq. Controls | Method | λ | OR (95% CI) | P value |
|---|---|---|---|---|---|---|---|---|---|---|---|---|
| Ancestry | Melanesian | **All** | **All** | **607** | **1229** | **1625** | **0.31** | **0.26** | **LMM** | **1.06** | **1.37 (1.19–1.57)** | **$1.2 \times 10^{-5}$** |
| | | iTaukei | All | 307 | 553 | 790 | 0.32 | 0.24 | LMM | 1.02 | 1.34 (1.07–1.67) | 0.011 |
| | | | Matched | 296 | 296 | 592 | 0.32 | 0.25 | LMM | 1.01 | 1.49 (1.15–1.94) | 0.003 |
| | | | | | | | | | LR | 1.01 | 1.51 (1.15–1.98) | 0.0024 |
| | | Kanak | All | 280 | 154 | 397 | 0.31 | 0.22 | LMM | 1.03 | 1.80 (1.30–2.49) | 0.00039 |
| | | | Matched | 153 | 153 | 306 | 0.29 | 0.19 | LMM | 1.09 | 1.77 (1.22–2.58) | 0.0028 |
| | | | | | | | | | LR | 1.07 | 1.80 (1.21–2.69) | 0.0028 |
| | Polynesian | **All** | **All** | **160** | **233** | **379** | **0.27** | **0.21** | **LMM** | **1.02** | **1.53 (1.12–2.10)** | **0.0072** |
| | | Samoan | All | 61 | 74 | 134 | 0.32 | 0.18 | LMM | 0.99 | 2.07 (1.23–3.50) | 0.0062 |
| | | | Matched | 55 | 55 | 110 | 0.35 | 0.18 | LMM | 0.99 | 2.16 (1.25–3.75) | 0.0061 |
| | | | | | | | | | LR | 1.04 | 2.24 (1.21–4.15) | 0.0066 |
| | Fijian Indian | **All** | **All** | **168** | **151** | **318** | **0.18** | **0.12** | **LMM** | **1** | **1.91 (1.18–3.10)** | **0.0082** |
| | | All | Matched | 142 | 142 | 284 | 0.18 | 0.11 | LMM | 1 | 1.99 (1.18–3.36) | 0.0096 |
| | | | | | | | | | LR | 1.00 | 2.02 (1.18–3.49) | 0.0092 |
| | Mixed and other | **All** | **All** | **71** | **236** | **218** | **0.26** | **0.17** | **LMM** | **1.02** | **1.54 (0.97–2.46)** | **0.069** |
| Country | Fiji Islands | All | All | 532 | 751 | 1245 | 0.27 | 0.22 | LMM | 1.02 | 1.39 (1.16–1.67) | 0.00043 |
| | New Caledonia | All | All | 422 | 362 | 779 | 0.28 | 0.18 | LMM | 1.02 | 1.57 (1.25–1.97) | $8.6 \times 10^{-5}$ |
| | Samoa | All | All | 61 | 75 | 135 | 0.32 | 0.18 | LMM | 0.99 | 2.10 (1.25–3.54) | 0.0054 |

CI, confidence interval; effective, effective sample size; freq., frequency; LMM, linear mixed model; LR, logistic regression; OR, odds ratio; RHD, rheumatic heart disease; Subpopn, subpopulation. Lines highlighted in bold refer to the initial discovery and replication analyses, while other lines refer to subsequent sensitivity analyses. The genomic control factor (λ) was calculated from a genome-wide analysis using the analytical method indicated.

**Structural consequences**. We next investigated the structural consequences of *IGHV4-61\*02*. Of the five nonsynonymous variants associated with the allele (Fig. 3c), only the proline to alanine at the IMGT (International Immunogenetics Information System) residue 46 is predicted to have a damaging effect on protein structure using the Polyphen-2 score (Supplementary Fig. 7c)[16]. Residue 46 is a component of the heavy chain interface framework loop (Fig. 3c) that has an important role in determining the orientation of the heavy chain variable domain relative to light chain variable domain[17], itself a key influence on the binding properties of the immunoglobulin molecule[17,18]. In comparison, there is limited evidence that the other four amino acid changes associated with *IGHV4-61\*02* impact on structure or function. Changes to the tyrosine residues at 55 and 58 fall adjacent to and within the second heavy chain complementarity determining region (CDR-H2) respectively, yet do not appear to alter the structure as they do not change the canonical class of the loop[19,20]. These residues may, however, affect binding without changing structure, particularly because tyrosine residues have high propensity to be in contact with antigen[21] and these positions often take part in binding[22]. The change from valine to isoleucine at residue 30 falls within the first heavy chain complementary determining region (CDR-H1), a position known to divide the first CDR into two loops[23], but there are insufficient structural data to establish the consequences of this change. Finally, the change from glutamic acid to glutamine at residue 17 is the least likely to affect structure because of the similar chemical properties of these amino acids and the fact that residue 17 lies on the surface of the protein, away from the binding site or the variable-heavy to variable-light domain interface.

## Discussion

In the first GWAS of RHD published to date, we identified a novel susceptibility signal in the IGH locus. While the relevance of these results outside Oceania remains to be assessed, the consistency of the signal across distinct ancestral groups and various sensitivity analyses and its correlation with diagnostic certainty adds weight to our findings.

Despite the fundamental role played by antibodies in adaptive immunity, germline variation in immunoglobulin genes has seldom been robustly connected to disease susceptibility[24]. Human immunoglobulin molecules are composed of heavy and light chains made up of constant and variable domains. During B-lymphocyte maturation, the heavy and light chain variable domains are generated through a process of recombination, junctional diversification and somatic hypermutation of the underlying gene segments[25]. The IGH locus is complex consisting of an estimated 123–129 variable (38–46 annotated as functional), 27 diversity (23 functional) and 9 joining (6 functional) gene segments[26,27]. Extensive structural variation and numerous short genetic variations introduce considerable diversity with a different number of functional variable gene segments present on each haplotype[24]. There is also substantial population stratification and it is highly likely that yet more variability will emerge as further complete haplotypes from diverse global populations are sequenced[26]. As in the HLA locus, the germline variation in the gene segments has been ordered into alleles, with two or more alleles defined for most of the heavy chain variable gene segments[14]. Crucially, although examples are scarce, this germline variation is thought to be an important determinant of antibody function as well as influencing the naive expressed repertoire[24] and consequently such variation has long been predicted to influence susceptibility to infectious and autoimmune disease[24].

In the candidate gene era, germline variation in variable gene segments was linked to susceptibility to a number of autoimmune diseases including multiple sclerosis, rheumatoid arthritis and systemic lupus erythematous, although the limited reproducibility of these results cast doubt on the validity of these associations[24]. Surprisingly, in the GWAS era, only two disease-focused studies— investigating Alzheimer's disease[28] and Kawasaki disease[29]—have reported findings at the IGH locus; however,

neither signal reached genome-wide significance nor localized to a specific gene segment. Indeed, the scarcity of GWAS findings at the IGH locus may be because this locus remains difficult to study. Key challenges include limited knowledge of IGH polymorphism, poor tagging by current standard genotyping arrays and deficiencies in the publicly available sequence data for this locus, much of which is derived from transformed B-lymphocytes that have typically lost components of the locus due to recombination[24]. The limitations of current genotyping arrays for study of the IGH locus are perhaps best illustrated by the fact that only 16 directly genotyped variants were included in our imputation scaffold from the entire 1,255 kb locus. Thus, although these variants effectively tagged the *IGHV4-61\*02* signal, it is highly likely that much of the remaining IGH polymorphism was poorly represented in our analysis, a problem that afflicts essentially all published GWASs to date[24]. The complexity of the IGH locus is further demonstrated by our discovery of a novel *IGHV4-61* allele that we speculate has arisen through a gene conversion event. Given the highly repetitive nature of the locus, it is plausible this is one of many such events, underscoring the need for further mapping of the locus to facilitate more accurate disease association studies. Moreover, particular effort will be needed to understand the diversity of IGH polymorphism in non-European populations[30], not least because these groups experience a disproportionate burden of infectious and inflammatory disease. Overall, however, our link between an *IGHV4-61* allele and RHD susceptibility may be an important step forward for understanding the immunogenetic determinants of autoimmune disease in general.

It has long been established that immunoglobulin deposits are an important feature of the pathology of RHD[31]. Interestingly, human hybridoma-derived immunoglobulins containing related heavy chain domains were previously shown to bind relevant streptococcal and host antigens including group A streptococcal carbohydrate and cardiac myosin[32]. In addition, autoantibodies against the same heavy chain domains were among 12 auto-antigens identified in sera from ARF patients screened using a human heart complementary DNA library[33]. At present, we conjecture that individuals who possess the *IGHV4-61\*02* allele are predisposed to produce autoreactive antibodies promoting valvulitis. Excitingly, knowing that a specific heavy chain gene segment contributes to susceptibility provides a potential route to identify relevant bacterial antigen(s) that could have important ramifications for the development of a much-needed GAS vaccine. Plausibly, such an antigen might itself be taken forward as a vaccine candidate, providing the theoretical risk of inducing autoimmunity by vaccination could be circumvented[34].

This study has two main limitations. First, by the standards of modern GWAS, our total sample size is relatively small, and hence it is likely many variants with smaller effects will go undetected until larger collections are assembled. Nonetheless, our study was well powered to detect the vast majority of large effect variants reported in the candidate gene era[8], especially those reported in HLA locus where signal in our study was negligible (minimum FE meta-analysis, $P = 0.0005$). Second, as we focused on variants with consistent direction and magnitude of effects across ancestral groups, our analysis provides little insight into variants with population-specific effects. As such, population-specific findings can provide important insights into biology, and this issue is worthy of further attention, perhaps by exploiting the underlying population genetics through techniques such as admixture mapping[35].

In summary, this first disease-focused Oceanian GWAS provides a new lead into the pathogenesis of RHD and mandates further research into the impact of germline IGH variants on susceptibility to RHD and potentially other autoimmune diseases.

## Methods

**Sample collections.** Genetic material was obtained with informed consent from cases and controls recruited to a number of distinct studies. Specifically, we established new collections from Fiji, New Caledonia and Samoa and we used samples from an existing collection covering Fiji, New Caledonia, Vanuatu, Samoa, Tonga, Cook Islands and French Polynesia (Fig. 1a). Cases of RHD were defined on the basis of: a history of valve surgery for RHD, a definite RHD diagnosis by echocardiography or borderline RHD diagnosis by echocardiography with prior ARF. All data pertaining to valve surgery, echocardiographic findings or histories of ARF were obtained from medical records. Echocardiographic diagnoses were based on criteria published by the World Heart Federation (WHF)[15] with a slight modification to the mitral stenosis definition so that it encompassed patients with a valve area of $-2 \, cm^2$ that is of equivalent diagnostic significance to the gradient $>4 \, mm \, Hg$ included in the WHF criteria[36]. Following the approach of the Wellcome Trust Case Control Consortium[37] and others, controls were members of the general population with limited or no phenotype information available. Summary characteristics for the cases are presented in Supplementary Fig. 1a.

*Fiji.* Children and adults with incident or prevalent RHD were recruited as cases between October 2012 and June 2014 from inpatients and outpatients at the Colonial War Memorial Hospital, Suva, Fiji, and at the Lautoka General Hospital, Lautoka, Fiji. Two pragmatic approaches were used to identify adult volunteers as controls: first, cases were requested to bring an unrelated friend or neighbour to clinic; second, adults were recruited during health promotion visits to communities in which cases were resident. The population of Fiji consists mostly of Oceanian peoples (including Indigenous iTaukei and migrant Polynesians) and Fijians of Indian decent (that is, South Asians), who emigrated from India in the 1900s, all of whom were eligible to take part. In total, DNA samples were obtained from 598 cases and 913 controls. Ethical approval was granted by the Fiji National Health Research Committee and the Fiji National Research Ethics Review Committee as well as the Oxford University Tropical Research Ethics Committee.

*New Caledonia.* Children and adults with incident or prevalent RHD were recruited as cases between March and December 2013 from inpatients and outpatients at the Hôpital de Gaston-Bourret, Nouméa, New Caledonia, and outpatients known to the Agence Sanitaire et Sociale de Nouvelle Calédonie, a government-funded public health service. Adult volunteers were recruited as controls pragmatically by requesting the case bring an unrelated friend or neighbour to clinic. The population of New Caledonia consists of Oceanian peoples (including Indigenous Kanak and migrant Polynesians), Europeans and East Asians, all of whom were eligible to take part. In total DNA samples were obtained from 492 cases and 365 controls. Ethical approval was granted by the Hospital Ethics Committee at the Hôpital de Gaston-Bourret and the Comité d'Evaluation Ethique de l'Inserm as well as the Oxford University Tropical Research Ethics Committee.

*Samoa.* Children with RHD were recruited during screening by the Rheumatic Rescue initiative between January and November 2014 undertaken in collaboration with the Samoa Ministry of Health. All those participating in the study reported Polynesian ancestry. In total, DNA samples were obtained from 70 cases with definite RHD according to the WHF criteria and 41 controls. In addition, DNA samples were available from 19 children with borderline RHD according to the WHF criteria and 44 children with mild nondiagnostic valve abnormalities. Although used for sensitivity analyses, both groups were excluded from the main analysis. Approval for the study was granted by the Samoa Ministry of Health as well as institutional review boards at Brigham Young University and Utah Valley University.

*Existing samples.* Additional samples were available from Oxford University studies in the Pacific region undertaken during the 1980s and 1990s[38–46]. These anonymized samples were originally collected for studies of haemoglobin genes and later the HLA locus but have subsequently been used for studies of various loci including, for example, the *CCR5* (ref. 47) and *HFE* genes[48]. Most samples were obtained from healthy adult volunteers but series of cord bloods were collected from consecutive healthy newborns at hospitals on the islands of Espiritu Santo and Maewo in Vanuatu[44] and Tahiti in French Polynesia[43]. Data from 658 samples from Vanuatu, 144 from Fiji, 32 from New Caledonia, 55 from Samoa, 49 from the Cook Islands, 33 from Tonga, and 84 from French Polynesia were used in this analysis. Permission for genetics research was granted at the time by various local and national institutions; permission to reuse samples for this study was granted by the Oxford University Tropical Research Ethics Committee.

**Preparation of DNA.** We obtained genetic material by sampling peripheral blood in Fiji and New Caledonia and by sampling saliva in Samoa. Blood samples collected in Fiji were stored in EDTA and frozen at $-80 \, °C$ until extraction. Blood samples collected in New Caledonia were stored in DNAgard (Biomatrica, USA) and kept at room temperature for up to 6 months. Saliva was collected using Oragene kits (DNA Genotek, Canada). DNA was extracted from blood collected in Fiji by an in-country research assistant using salt precipitation and from blood collected in New Caledonia after shipment to the United Kingdom by LGC Limited (UK). DNA was extracted from saliva using DNA Genotek proprietary kit by research assistants at the Brigham Young University. Other samples had previously been extracted using standard approaches. Extracted DNA from Fiji and New Caledonia was prepared for analysis at LGC Limited (UK) where quantification

was performed by ultraviolet spectrophotometry. Because 483 samples from Fiji were of insufficient concentration for genome-wide genotyping, they were whole-genome amplified using LGC Limited proprietary primer-extension pre-amplification PCR. DNA from other collections was quantified and prepared for analysis by a research assistant at the University of Oxford. Quantification at the University of Oxford was performed using the PicoGreen (Life Technologies, USA) reaction.

**Genome-wide genotyping and quality control.** We genotyped the complete collection of 3,412 DNA samples at the Oxford Genomics Centre at ∼300,000 variants using the HumanCore-24 BeadChip (Illumina Inc., USA). After calling using the default settings of the clustering algorithm implemented in GenomeStudio software (Illumina Inc.), the data set was aligned to the forward strand of the Genome Reference Consortium Human Build 37 as previously described (http://www.well.ox.ac.uk/∼wrayner/strand).

We employed standard approaches to quality control (QC) the genotyping data[49] with most steps performed using PLINK software version 1.90 (beta)[50]; we did not perform sex checks because information on phenotypic sex was incomplete. Starting with 'per individual' QC (Supplementary Fig. 1b), we measured missingness in each sample and examined its relationship with autosomal heterozygosity (Supplementary Fig. 4a). Based on this relationship, we removed genome-wide amplified samples with missingness > 5% and other samples with missingness > 1%. In addition, we removed samples with inbreeding coefficient ($F$) > 0.227 (the mean plus three s.d. values, of the individuals reporting Melanesian or Polynesian ancestry) or < − 0.361 (the mean minus three s.d. values of the individuals reporting Fijian Indian, mixed or other ancestry). Finally, we removed 14 duplicates with a cutoff of identity by descent > 0.90 measured in PLINK.

We then performed 'per variant' QC (Supplementary Fig. 1c). The overall genotyping rate was high at 99.3% and only 4,308 variants had missingness > 2%. We removed all variants with minor allele frequency (MAF) < 1.25% because such variants are usually less reliably genotyped[49]. We kept variants with MAF 1.25 to 5% but applied stricter missingness thresholds (Supplementary Fig. 1c). Finally, we removed variants with extreme deviation from Hardy–Weinberg equilibrium using a previously suggested threshold of variants with Hardy–Weinberg equilibrium $P$ values < $10^{-50}$ (ref. 51).

**Population-specific imputation panel.** Oceanian populations are not represented in current reference panel data widely used for imputation. To remedy this, we whole-genome sequenced 64 samples from New Caledonia targeting four times (4 ×) coverage (Supplementary Fig. 2a). In addition, because higher density array data improve the accuracy of phasing[10], we genome-wide genotyped these same 64 samples from New Caledonia along with 64 samples from the Fiji study using the denser HumanOmniExpressExome-8 BeadChip (Illumina Inc.) that includes ∼960,000 variants of which 273,000 are exonic. Both sets comprised equal numbers of young cases with severe disease and older controls known to be asymptomatic randomly selected from Melanesian participants thought least likely to show European admixture: Kanak individuals from Province Nord on Grand Terre for New Caledonia and iTaukei individuals from rural parts of the Central Division on Vitu Levu for Fiji.

HumanOmniExpressExome-8 genotyping was performed as described above for the HumanCore-24 data with identical QC procedures. Sequencing by synthesis was performed at the Oxford Genomics Centre using the HiSeq 2500 System (Illumina Inc.) and the TruSeq DNA PCR-Free Library Preparation kit (Illumina Inc.). Reads were mapped to Build 37 using Stampy software[52] version 1.0.25 before deduplication, local realignment and base score recalibration using the Genome Analysis Toolkit (GATK) software[53] version 3.3. We then called SNPs and INDELs with phred-scaled confidence > 30.0 using GATK HaplotypeCaller[54]. Once called, the sequenced data were phased on to the genotyped data as previously described[10] using SHAPEIT software[55] version 2.5.

**Genome-wide imputation.** Because prephasing reduces the computation burden of imputation without reducing accuracy, we prephased the 239,990 HumanCore-24 variants that had passed QC in the 3,234 individuals who had passed QC using SHAPEIT. We then performed genome-wide imputation using IMPUTE2 software[56,57] with the 'merge_ref_panel' option to integrate the Oceanian sequence data with the 1000 Genomes panel. To assess whether using the integrated data improved accuracy, we undertook the chromosome 1 analysis with and without the Oceanian sequence data and examined concordance (Supplementary Fig. 2d).

**Assessing relatedness.** In the 3,234 quality-controlled individuals, we estimated relatedness using RelateAdmix software[58] version 1.0 that provides more accurate estimates of relatedness in the presence of admixture than standard tools[58]. Admixture estimates (Fig. 1c) were made using a model-based clustering algorithm implemented in fastSTRUCTURE software[59] version 1.0. Altogether, we uncovered a high degree of relatedness (Supplementary Fig. 4b), especially in comparison to standard population-based case–control association analyses[49]. Accordingly, to minimize the effect of such relatedness on the analysis, especially in the presence of marked population structure (see next section), we removed one individual from each related pair of first- or second-degree relatives in succession until no such relationships remained, necessitating the removal of 382 individuals

(Supplementary Fig. 1b). We used a cutoff of relatedness ($r$) > 0.1875 that lies midway between the theoretical relatedness of second- and third-degree relatives[49].

**Genomic ancestry and stratification.** We performed principal component (PC) analysis (Supplementary Fig. 3) using the tool implemented in Genome-wide Complex Trait Analysis (GCTA) software[60] version 1.24.4 by combining our data set with selected individuals from the Human Genome Diversity Project panel[61]. To investigate the effects of population structure on the association analyses, we performed genome-wide association analyses using either logistic regression or linear mixed models (described below), plotting the negative common logarithm of the resulting $P$ values on quantile–quantile plots using the R package 'qqman' that also permitted estimation of the genomic control factor ($λ$)[62]. In preliminary analyses, the ancestral heterogeneity caused considerable inflation of the distribution of the test statistics, even limiting the analysis to individuals from a single country or single ancestral group (logistic regression, $λ = 1.54–5.03$). To counter this problem, therefore, the analysis was stratified by ancestry based on the four clusters detected in the PC analysis. We defined these clusters pragmatically by selecting individuals < 2 s.d. from the mean of the first PC and < 3 s.d. from the mean of the second and third PCs for their self-reported ancestry (Supplementary Fig. 3a–d).

Within each strata, however, there remained significant evidence of structure that—reflecting the amalgamation of iTaukei individuals from Fiji, Kanak individuals from New Caledonia and Ni-Vanuatu individuals from Vanuatu—was especially apparent in the Melanesian stratum (Supplementary Fig. 3e). For sensitivity analyses, therefore, we generated four subsets of matched case–control pairs made up of individuals reporting iTaukei ancestry from the Melanesian strata, Kanak ancestry from the Melanesian strata, Samoan ancestry from the Polynesian strata or Fijian Indian ancestry from the Fijian Indian strata (Supplementary Fig. 3f–i). To achieve this, based on a method described previously[63], we weighted the first 20 PCs by how much phenotypic variance each PC explained in multiple regression. We then calculated the Euclidean distance between all individuals and optimally matched each case to the single nearest control using the R package Optmatch[64].

**Association testing.** Our primary measure of association between phenotype at single loci employed in the GWAS analyses was the LMM, also termed the variance components model. This model explicitly accounts for correlations in phenotypes due to relatedness, thereby minimizing confounding due to population structure, admixture and cryptic relatedness[65]. More specifically, we used GCTA to calculate kinship matrices in each ancestral stratum using a leave-one-chromosome-out approach in which the kinship matrix for a given chromosome is calculated using all directly genotyped variants on the remaining 21 autosomes with MAF > 1.25% in that strata[60]. We then, for each genotyped and imputed autosomal variant, used linear regression to model the relationship of a dependent variable, representing case–control status, with independent variables, representing the dose of the minor allele at the variant of interest, estimated by imputation (fitted as a fixed effect) and genome-wide structure and relatedness calculated by decomposition of the kinship matrix (fitted as a random effect). We converted estimates of effect size and standard errors from LMM to odds ratios and confidence intervals by linear transformation[60]. For sensitivity analyses we also used logistic regression models implemented in SNPTEST software[66] version 2.5.1. Throughout we used accepted thresholds for genome-wide significance ($P < 5 × 10^{-8}$) and suggestive significance ($P < 1 × 10^{-5}$)[67]. At this level, with a total sample size of 1,006 cases and 1,846 controls, we achieved our aim of 80% power to detect variants with effect size of 1.5 or more at MAF > 20%. Finally, to aid interpretation, we calculated effective sample size that provides an indication of the sample size had there been an equal number of cases and controls. Based on the ratio of the number of samples, effective sample size for a case–control study is, $N\_eff = 4/((1/N\_cases) + (1/N\_controls))$.

**Meta-analysis.** Having undertaken the discovery analysis, we combined the association statistics genome-wide with those from the three remaining ancestral strata using FE meta-analysis. Despite the requirement for no significant heterogeneity, FE meta-analysis remains the method of choice for discovery analyses because random effects meta-analysis is markedly conservative in the presence of heterogeneity[12]. Genome-wide meta-analysis was performed using inverse variance weighting as implemented in METASOFT software[68] version 2.0.1. In addition, for fine-mapping, we used a Bayesian meta-analysis technique that explicitly accounts for heterogeneity between ancestral groups using estimates of divergence such as $FST$ as a prior[11]. A Bayes factor (BF) measures evidence in favour of association and if the common logarithm of the BF exceeds 6.0, a variant is considered to have reached genome-wide significance[69]. Assuming a single causal variant at each locus, the posterior probability that the $j$th variant is causal can be estimated as $φ_j = BF_j / \sum_k BF_k$ where $\sum_k BF_k$ is the sum of the BFs for all variants included in the analysis in a locus extending 500 kb either side of the lead variant[69]. A 99% credible set can then be defined by ranking variants until their cumulative posterior probability exceeds 0.99 (ref. 13).

**Sanger sequencing.** We pragmatically selected a portion of samples for further analysis at the *IGHV4-61* locus based on the ready availability of genetic material. Using PCR, we first amplified 1,599 bases on chromosome 14 containing the *IGHV4-61* gene segment with primers (Supplementary Fig. 8a) designed such that they were specific to this locus using the NCBI Primer Blast tool[70] and optimized with respect to annealing temperature, extension time and concentrations of MgCl$_2$, dimethylsulfoxide and template (Supplementary Fig. 8b). However, to compensate for the presence of a common SNP (rs11621753) within the binding site of the forward primer, we substituted the corresponding base on the forward primer (position 18) for the genotype of the alternate allele (that is, C to A substitution) because we had found in preliminary work (by examining the genotypes of variants in linkage disequilibrium with rs11621753 in sequenced samples) that in heterozygous individuals under stringent conditions, primers matching the reference sequence amplified only chromosomes carrying the reference allele, whereas primers matching the alternate allele amplified chromosomes carrying both reference and alternate alleles. We visualized both products by gel electrophoresis and only those samples that had successfully amplified were taken forward for sequencing. Because relatively few of the samples from Samoa amplified (likely reflecting collection in saliva and/or degradation), we found it necessary in 42 samples to perform an additional round of PCR using a nested approach, amplifying the 473 bp product of the sequencing primers using the 1,599 bp product of the initial PCR as the template (Supplementary Fig. 8a). Having originally intended to sequence 10–15% samples from the wider study, we successfully took forward 364 samples (12.7%) for sequencing providing a broadly representative subset of the collection: iTaukei Fijian from Fiji ($n = 83$), Kanak ($n = 135$), Samoan ($n = 61$) and Fijian Indian ($n = 85$). Sequencing reactions were carried out in 10 µl of cleaned-up PCR product using BigDye Terminator reagents (Applied Biosystems, USA). We used separate reactions for forward and reverse strand primers targeting a 473 bp product covering all but the last 42 bases of the second exon of *IGHV4-61*. The sequencing reactions were then cleaned up by ethanol/EDTA/sodium acetate precipitation. Sequencing was performed at the University of Oxford Department of Zoology using a standard ABI Prism 3730xl DNA Analyser (Applied Biosystems, USA). Two authors (K.A. and A.R.) read the sequences for the two key variants (rs202117805, rs539138682). Where there was discrepancy, as happened in only 4 of 528 calls (0.76%), a third author was consulted (A.J.M.) and agreement reached. One author (K.A.) read the sequences for a further seven variants (rs2516897, rs201453364, rs2072046, rs202166511, rs200931578, rs201691548 and rs201076896). All three were blinded to imputed genotypes and case–control status. Finally, to reimpute this region into the wider data set, the 9 chain-termination genotypes for the 364 sequenced individuals were first phased using SHAPEIT[55] with 19 other variants within 250 kb of *IGHV4-61* that had been either directly genotyped or imputed with high confidence (missing information <1%). Then, with the 19 other variants in the *IGHV4-61* locus providing a scaffold, we imputed using IMPUTE2 (refs 56,57) the 9 chain-termination genotypes for the 3,234 individuals who had passed QC using the genotypes. As recommended for follow-up analysis of putative disease-associated loci, this local imputation was performed without prephasing[66].

**Data availability.** Genotype and phenotype data underlying the manuscript have been deposited in the European Genome-phenome Archive under accession number EGAS00001001881. Some restrictions on access and usage apply with much of the data set restricted to research focused on RHD. Access to certain components of the data set requires regulatory approval from the country where the samples were obtained. Further information about access to the data set is provided at http://www.rhdgenetics.net/pacific.html where an elemental data set sufficient to reproduce the *IGHV4-61*02 signal reported here is available for immediate download. The novel *IGHV4-61* allele provisionally designated *IGHV4-61*09 has been deposited in GenBank under accession number KX389267.

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

## Acknowledgements

This research was supported by grants awarded to T.P. from the British Heart Foundation (PG/14/26/30509), the Medical Research Council (G1100449) and the British Medical Association (Josephine Lansdell Grant 2012). In addition, M.M.M. received funding from La Fondation pour la Recherche Médicale (FDM20140630267), la Fédération Française de Cardiologie (Bourse d'études à l'étranger) and la Fondation Lefoulon Delalande (Bourse post-doctorale); J.N. and C.M.D. received funding from the Engineering and Physical Sciences Research Council (EP/G037280/1); A.J.M. holds a Wellcome Trust Clinical Research Training Fellowship (106289/Z/14/Z); A.C.S. holds a Career Development Fellowship from National Health and Medical Research Council of Australia (1127077) and a Future Leader Fellowship from the National Heart Foundation of Australia (101174); and A.V.S.H. holds Senior Investigator awards from the Wellcome Trust (104750/Z/14/Z) and National Institute for Health Research (NF-SI-0514-10158). None of these funders had any role in study design, data collection and analysis, decision to publish or preparation of the manuscript. We thank the High-Throughput Genomics Group at the Wellcome Trust Centre for Human Genetics for generating the genotyping and sequencing data, subsidized by a core award from the Wellcome Trust (090532/Z/09/Z). We also thank Dr Tara Mills for valuable suggestions regarding the sample collection, Professors Gilean McVean, Jonathan Marchini and Andrew Morris for helpful advice about study design and statistical analysis and Dr Corey Watson for useful discussions concerning the immunoglobulin heavy chain locus. Finally, we thank Professor John Clegg for permission to use the existing Oceanian sample collection and Professors David Weatherall, Don Bowden and their many colleagues for the work involved in establishing that collection.

## Author contributions

T.P., M.M.M., J.K., A.R., E.M., X.J., M.L.P., T.C., A.C.S. and A.V.S.H. organised and designed the study. T.P., M.M.M., A.C.S. and A.V.S.H. managed the study. T.P., M.M.M., J.K., M.L.P., T.C., J.B.A., H.T., L.A., M.A., C.B., S.M.C., A.J., M.A.K., R.M., M.N., S.N., T.N., B.N., N.S. and B.W. recruited the patients. T.P., K.A., A.R., J.K.K., K.J.R., R.K., and W.J.M. did the laboratory studies. T.P., K.A., J.N., A.R., A.J.M., C.M.D. did the statistics and bioinformatics. T.P., M.M.M., J.K., K.A., J.N., A.R., A.J.M., J.K.K., C.M.D., A.C.S. and A.V.S.H. contributed to the interpretation of the results. T.P. wrote the first draft of the manuscript under the supervision of A.C.S. and A.V.S.H. All authors contributed to revisions and approved the final version for publication.

## Additional information

**Competing interests:** The authors declare no competing financial interests.

## Pacific Islands Rheumatic Heart Disease Genetics Network

Lori Allen[13], Marvin Allen[14], Corinne Braunstein[15], Samantha M. Colquhoun[11], Aurélia Jewine[15], Maureen Ah Kee[7], Rina Kumar[16], William John Martin[17], Reapi Mataika[3], Marie Nadra[15], Shahin Nadu[18], Take Naseri[19], Baptiste Noël[15], Nathalie Simon[15] & Brenton Ward[12]

[13]Department of Public & Community Health, 987 South Geneva Road, Utah Valley University, Orem, Utah 84058, USA. [14]Department of Cardiology, Central Utah Clinic, 1055 North 500 West Street, Utah 84604, USA. [15]Department of Cardiology, Centre Hospitalier Territorial de Nouvelle-Calédonie, 7, avenue Paul Doumer, 98849 Nouméa, New Caledonia. [16]Centre for Communicable Disease Control, Ministry of Health and Medical Services, Princess Road, Suva, Fiji. [17]Walter and Eliza Hall Institute, 1G Royal Parade, Parkville, Melbourne, Victoria 3052, Australia. [18]Department of Cardiology, Ministry of Health and Medical Services, Colonial War Memorial Hospital, Brown Street, Suva, Fiji. [19]Office of the Director General, Ministry of Health, Moto'otua, Ifiifi Street, Apia, Samoa.

