## [Peer Review File · Nature Communications]

Reviewers' comments:

Reviewer #1 (Remarks to the Author):

This was an interesting and informative manuscript, describing a carefully-conducted genome-wide association study (GWAS), the first to be carried out in rheumatic heart disease (RHD). Overall the results were quite compelling, although they will require investigation in other population cohorts.

I have only minor comments:

Page 3 line 12: "ARF" needs to be defined before its first usage.

Page 3 line 26: With only 239,990 variants typed on the Illumina HumanCore platform, this seems like a remarkably low-density GWAS chip. Are we supposed to conclude from Suppl Fig 2D that it is, in fact, sufficiently dense to capture the overall variation (and therefore to be used for imputation)?

Page 4 line 24: Figure 2 does not really illustrate the fact that there was a signal in the discovery cohort that "replicated" in the other cohorts. (This is illustrated better in Table 1 and Fig 3b). I would like to see the Manhattan plot from the discovery (Melanesian) cohort (perhaps as supplementary material) as well as the currently-shown Manhattan plot from the overall final meta-analysis.

Page 5 line 12: By "Table" do you mean "Table 1"?

Supplementary Fig 4A: Can you explain why people seem to fall into 2 separate groups (divided at the 0.5% vertical line) with respect to their "missingness"?

Reviewer #2 (Remarks to the Author):

Review of Parks et al. RHD and IGH

This is a significant paper that is highly suitable for Nature Communications. It presents an important result regarding genetic associations with Rheumatic Heart Disease. Since this association was found in the IGH region (immunoglobulin heavy chain), this might be indicative of many undiscovered disease associations in this complicated genomic region, which gives the findings of this paper general application and significance.

I am much more familiar with the genomics of IGH than I am of the sophisticated mapping/association statistics that are presented in this manuscript, so my review will be primarily focused on IGH.

Major comments:

Abstract. The authors make two important points. 1. There is a strong association for RHD in IGH. 2. This is one of the few GWAS studies that have shown an association between diseases that we suspect would map to IGH (either a priori due to production of antibodies, or due to earlier associations observed using coarser mapping approaches). But the last sentence of the Abstract combines these two ideas and I find that confusing; I think it should be rewritten to make those two points more separate and highlighted.

The proposed model is that all of the sequences that are being analyzed come from the gene/locus

VH4-61. And the authors do a careful job of describing how they optimized that the sequences came from 4-61. However, the authors suggest that there has been a 2 codon deletion that is equivalent to a deletion in the gene IGHV4-59, and that this has independently occurred at the 4-61 locus and produced a new allele of 4-61 provisionally called 4-61*09. The authors do mention the high complexity and repetitive nature of this locus. Given that, I think it is impossible to rule out the possibility that their primers were amplifying from 4-59 to get produce this allele. At the minimum, I think the authors have to refer to this possibility. I think the authors should also mention that the new allele proposed at 4-61 could be a gene conversion event from an allele at 4-59? I realize that there is a constraint on the size of the main part of the manuscript. Perhaps this can all be done in supplementary discussion, to not take space away from the main message of an association of RHD with 4-61*02, but the message is incomplete without this extra information. The authors properly state how complicated IGH is, and how this might make it difficult to properly genotype at IGH and properly test for associations with important, common immune-function diseases; but without at least a mention of other possible explanations for this new allele, 4-61*09, I think their narrative is incomplete.

In a similar manner, again for a "teaching moment", I think the authors should just mention the density of SNPs in IGH on their GWAS chips that were used. I think the 300K Human core platform has about 100 SNPs/Probes in IGH V genes, so there probably is enough to capture a strong association such as they observed. But I think the whole enterprise would be greatly enhanced if they brought this up as a point; i.e., some GWAS studies (such as those based on the immunochip, which I think only has 10 probes in IGH) just couldn't possibly map even a strong association because of low density of probes in IGH. I don't think this would take much extra space.

p. 3 line 5. This is an extremely important point, that perhaps really hasn't been pointed out that often. Lots of these diseases are only important in regions with insufficient access to health care, and that is exactly where our knowledge of germline polymorphisms is the worst. Perhaps make this even more explicit?

Confirmation by Sanger Sequencing Section – The authors present some data on the allelic variants, from both electropherograms, and in tables of SNPs/amino acid changes. I think it would be very important and helpful to present the alleles being examined, 4-61 *02, *09, the 4-59 allele with the deletion, as a full lined up sequence. Best to follow the IMGT Facts Book format. Again, this could be in a supplemental figure, but it would really help people see the nucleotide and amino acid changes.

More minor/editorial points:

p. 2 line 6 – isn't a comma necessary after analyzed?

p. 2 line 9 – I think the use of "missense" is confusing here. I would suggest amino acid changing. At least refer to these as missense, stating compared to *01 allele? In fact I think for these same changes, the authors use the term amino acid changes p. 5 line 31, so I would adopt "amino acid changing", or maybe best "non-synonymous"?

p. 3 line 12 define ARF?

p. 5 line 12 number of table got dropped

p. 5 line 28 "these" instead of "there"

p. 6 lines 26-30. This section reports on the number of V D and J genes. It is important to point out

that we do know the number of V genes for most haplotypes; it is highly variable between haplotypes. Important to mention Watson et al. 2012 here, and the fact that there are 2800 SNPs reported between Matsuda and CH17 full haplotypes, and multiple insertion and deletions reported in Watson et al. 2012. etc. Just a few sentences at most would complete this idea, giving some idea of what is known about the variability.

p. 11 line 1 "oceanic" got put in from autocorrect I assume?

p. 15 line 15/16. The authors removed samples with high "missingness." Given the lack of knowledge of germline polymorphisms in these under-studied populations, could the removal of samples due to missingness be non-random with respect to population? Could some of these populations have higher missingness not due to bad DNA preparation, but due to being less well-studied populations, and could this affect the results in any way?

Also, it is reported that samples with high inbreeding coefficients F were removed, but F has to be calculated in pairs of samples? Doesn't that need to be explained here?

p. 18 line 13 "the main paper" - what does that mean here?

p. 20 line 3 remove second "for further analysis"

REVIEWERS' COMMENTS:

Reviewer #2 (Remarks to the Author):

It appears that the authors carefully considered and responded to the reviewers' comments. I therefore feel the manuscript is ready for publication.

Response to Reviewer 1:

I have only minor comments:

Page 3 line 12: "ARF" needs to be defined before its first usage.

This has been corrected.

Page 3 line 26: With only 239,990 variants typed on the Illumina HumanCore platform, this seems like a remarkably low-density GWAS chip. Are we supposed to conclude from Suppl Fig 2D that it is, in fact, sufficiently dense to capture the overall variation (and therefore to be used for imputation)?

When we designed the project we selected an economical genotyping platform that would allow us to acquire an imputation scaffold in a large number of samples. Focusing on common variation, we supposed that fewer variants would be needed because linkage disequilibrium in Oceanian populations stretches over greater distances than in any other human population. Supporting this strategy, Suppl Fig 2D suggests that, using the population-specific low-coverage sequence data, we achieve reasonable imputation accuracy across the allele frequency spectrum. Interestingly, since our study was designed, it has been estimated that the HumanCore platform captures more than 75% of common variation in European, Asian and American populations with over 90% accuracy (1).

1. Nelson SC, Doheny KF, Pugh EW, *et al.* Imputation-based genomic coverage assessments of current human genotyping arrays. *G3 (Bethesda)* 2013; 3: 1795–807.

Page 4 line 24: Figure 2 does not really illustrate the fact that there was a signal in the discovery cohort that "replicated" in the other cohorts. (This is illustrated better in Table 1 and Fig 3b). I would like to see the Manhattan plot from the discovery (Melanesian) cohort (perhaps as supplementary material) as well as the currently-shown Manhattan plot from the overall final meta-analysis.

We have added this as Suppl. Fig. 5.

Page 5 line 12: By "Table" do you mean "Table 1"?

This has been corrected.

Supplementary Fig 4A: Can you explain why people seem to fall into 2 separate groups (divided at the 0.5% vertical line) with respect to their "missingness"?

The samples separate out into those genotyped from genomic DNA (GEN, open-circles, n= 2929) and those genotyped from genome-wide amplified DNA (AMP, open-squares, n=483). This separation is expected and relatively inconsequential because both cases and controls were amplified. However, different missingness thresholds were applied to the genomic and amplified samples at 0.5% and 5% respectively.

Response to Reviewer 2:

Major comments:

Abstract. The authors make two important points. 1. There is a strong association for RHD in IGH. 2. This is one of the few GWAS studies that have shown an association between diseases that we suspect would map to IGH (either a priori due to production of antibodies, or due to earlier associations observed using coarser mapping approaches). But the last sentence of the Abstract combines these two ideas and I find that confusing; I think it should be rewritten to make those two points more separate and highlighted.

Due to the word-limit constraints of Nature Communications it has been necessary to shorten our abstract. In the revised version only the latter of these points remains. Instead we have moved the original text to the last paragraph, where a brief summary of the results and conclusions are required. As suggested the two points are separated:

“Set in populations hitherto largely overlooked by genetics research, our study is the first GWAS of RHD to be reported, providing much needed insight into the pathogenesis of this devastating disease. Additionally, as the first study in the GWAS era to link germline coding variants in the IGH locus to disease susceptibility, our study suggests further should consideration be given to the role of IGH polymorphism in autoimmune disease.”

The proposed model is that all of the sequences that are being analyzed come from the gene/locus VH4-61. And the authors do a careful job of describing how they optimized that the sequences came from 4-61. However, the authors suggest that there has been a 2 codon deletion that is equivalent to a deletion in the gene IGHV4-59, and that this has independently occurred at the 4-61 locus and produced a new allele of 4-61 provisionally called 4-61*09. The authors do mention the high complexity and repetitive nature of this locus. Given that, I think it is impossible to rule out the possibility that their primers were amplifying from 4-59 to get produce this allele. At the minimum, I think the authors have to refer to this possibility. I think the authors should also mention that the new allele proposed at 4-61 could be a gene conversion event from an allele at 4-59? I realize that there is a constraint on the size of the main part of the manuscript. Perhaps this can all be done in supplementary discussion, to not take space away from the main message of an association of RHD with 4-61*02, but the message is incomplete without this extra information. The authors properly state how complicated IGH is, and how this might make it difficult to properly genotype at IGH and properly test for associations with important, common immune-function diseases; but without at least a mention of other possible explanations for this new allele, 4-61*09, I think their narrative is incomplete.

We have added Suppl. Note 1 and Suppl. Fig. 11 showing that in a subset of individuals the 1kb of sequence surrounding the novel allele better matches the IGHV4-61 locus than the IGHV4-59 locus. This is referred to from the results section with the sentence:

“Although the complexity of the IGH locus makes it difficult to be certain, it seems most likely that this novel allele has been amplified from the *IGHV4-61* locus rather than the *IGHV4-59* locus because the sequence surrounding *IGHV4-61*09* matched the former better than the latter (Supplementary Note 1).”

Wellcome Trust Centre for Human Genetics, Roosevelt Drive, Oxford. OX3 7BN
tomparks@well.ox.ac.uk • +44 7795 082724 • +679 8665994

We have also added text to the discussion further emphasizing the complexity and raised the possibility of a gene conversion event:

“The complexity of the IGH locus is further demonstrated by our discovery of a novel *IGHV4-61* allele, which we speculate has arisen through a gene conversion event. Given the highly repetitive nature of the locus, it is plausible this is one of many such events, underscoring the need for further mapping of the locus to facilitate more accurate disease association studies.”

In a similar manner, again for a “teaching moment”, I think the authors should just mention the density of SNPs in IGH on their GWAS chips that were used. I think the 300K Human core platform has about 100 SNPs/Probes in IGH V genes, so there probably is enough to capture a strong association such as they observed. But I think the whole enterprise would be greatly enhanced if they brought this up as a point; i.e., some GWAS studies (such as those based on the immunochip, which I think only has 10 probes in IGH) just couldn’t possibly map even a strong association because of low density of probes in IGH. I don’t think this would take much extra space.

The density of IGH variants on the HumanCore platform is also surprisingly low with only 16 variants directly genotyped variants included in our imputation scaffold. We have added two sentences to the discussion to highlight this issue:

“The limitations of current genotyping arrays for study of the IGH locus are perhaps best illustrated by the fact that only 16 directly genotyped variants were included in our imputation scaffold from the entire 1255kb locus. Thus, although these variants effectively tagged the *IGHV4-61*02* signal, it is highly likely that much of the remaining IGH polymorphism was poorly represented in our analysis, a problem that afflicts essentially all published GWAS to date²⁴.”

p. 3 line 5. This is an extremely important point, that perhaps really hasn’t been pointed out that often. Lots of these diseases are only important in regions with insufficient access to health care, and that is exactly where our knowledge of germline polymorphisms is the worst. Perhaps make this even more explicit?

We have added an additional sentence to the same section of the discussion further emphasizing this point:

“Moreover, particular effort will be needed to understand the diversity of IGH polymorphism in non-European populations³⁰, not least because these groups experience a disproportionate burden of infectious and inflammatory disease.”

Confirmation by Sanger Sequencing Section – The authors present some data on the allelic variants, from both electropherograms, and in tables of SNPs/amino acid changes. I think it would be very important and helpful to present the alleles being examined, 4-61 *02, *09, the 4-59 allele with the deletion, as a full lined up sequence. Best to follow the IMGT Facts Book format. Again, this could be in a supplemental figure, but it would really help people see the nucleotide and amino acid changes.

We have now added nucleotide and amino acid sequences in Suppl. Fig. 10.

More minor/editorial points:

p. 2 line 6 – isn't a comma necessary after analyzed?

We think not because it is the “genotyped and imputed variants” that are being analyzed.

p. 2 line 9 – I think the use of “missense” is confusing here. I would suggest amino acid changing. At least refer to these as missense, stating compared to *01 allele? In fact I think for these same changes, the authors use the term amino acid changes p. 5 line 31, so I would adopt “amino acid changing”, or maybe best “non-synonymous”?

For consistency we have now used the term non-synonymous throughout the paper.

p. 3 line 12 define ARF?

This has been corrected.

p. 5 line 12 number of table got dropped

This has been corrected.

p. 5 line 28 “these” instead of “there”

Although we recognise both would work we have a preference for “there being nil, marginal and significant effect respectively” over “these being nil, marginal and significant respectively”.

p. 6 lines 26-30. This section reports on the number of V D and J genes. It is important to point out that we do know the number of V genes for most haplotypes; it is highly variable between haplotypes. Important to mention Watson et al. 2012 here, and the fact that there are 2800 SNPs reported between Matsuda and CH17 full haplotypes, and multiple insertion and deletions reported in Watson et al. 2012. etc. Just a few sentences at most would complete this idea, giving some idea of what is known about the variability.

This section has been expanded and now includes a citation of the Watson et al. paper that describes the CH17 haplotype:

“The IGH locus is complex consisting of an estimated 123-129 variable (38-46 annotated as functional), 27 diversity (23 functional) and 9 joining (6 functional) gene segments^{26,27}. Extensive structural variation and numerous short genetic variations introduces considerable diversity with a different number of functional variable gene segments present on each haplotype²⁴. There is also substantial population stratification and it is highly likely that yet more variability will emerge as further complete haplotypes from diverse global populations are sequenced²⁶.”

p. 11 line 1 “oceanic” got put in from autocorrect I assume?

This has been corrected.

p. 15 line 15/16. The authors removed samples with high “missingness.” Given the lack of knowledge of germline polymorphisms in these under-studied populations, could the removal of samples due to missingness be non-random with respect to population? Could some of these populations have higher missingness not due to bad DNA preparation, but due to being less well-studied populations, and could this affect the results in any way?

Genome-wide missingness was calculated for each sample using variants across the frequency spectrum. Allowing for the samples that underwent genome-wide amplification, missingness was not related to ancestry, as illustrated in Suppl. Fig. 4A.

Also, it is reported that samples with high inbreeding coefficients F were removed, but F has to be calculated in pairs of samples? Doesn't that need to be explained here?

The inbreeding coefficient was calculated using PLINK software using the ratio of observed to expected homozygous genotypes genome-wide. This is the standard approach to assessing homozygosity for quality control purposes in genome-wide association studies.

p. 18 line 13 “the main paper” – what does that mean here?

This has been deleted for clarity.

p. 20 line 3 remove second “for further analysis”

This has been corrected.